# Diet-Related Attitudes, Beliefs, and Well-Being in Adolescents with a Vegetarian Lifestyle

**DOI:** 10.3390/healthcare11212885

**Published:** 2023-11-02

**Authors:** Loredana Benedetto, Ilenia Sabato, Carola Costanza, Antonella Gagliano, Eva Germanò, Luigi Vetri, Michele Roccella, Lucia Parisi, Costanza Scaffidi Abbate, Massimo Ingrassia

**Affiliations:** 1Department of Clinical and Experimental Medicine, University of Messina, 98125 Messina, Italy; ilenia.sabato21@gmail.com (I.S.); massimo.ingrassia@unime.it (M.I.); 2Department of Sciences for Health Promotion and Mother and Child Care “G. D’Alessandro”, University of Palermo, 90128 Palermo, Italy; carola.costanza@unipa.it; 3Department of Human and Pediatric Pathology “Gaetano Barresi”, University of Messina, 98125 Messina, Italy; antonellagagliano.npi@gmail.com (A.G.); eva.germano@unime.it (E.G.); 4Oasi Research Institute-IRCCS, Via Conte Ruggero 73, 94018 Troina, Italy; 5Department of Psychology, Educational Science and Human Movement, University of Palermo, 90128 Palermo, Italy; michele.roccella@unipa.it (M.R.); lucia.parisi@unipa.it (L.P.); costanza.scaffidi@unipa.it (C.S.A.)

**Keywords:** vegetarian, food choice, public health, adolescents, psychological well-being

## Abstract

Vegetarianism can meet healthy, ethical, or ecological values (such as equality and protection of animals or the environment). At the same time, it can represent a response to the need for self-determination in adolescence. Furthermore, some studies show vegetarians have greater depressive risk and a lower sense of body satisfaction. Considering the spread of non-meat diets in the Western world, researchers have investigated the benefits and risks to physical and psychological health. Despite this, few studies have been conducted on factors influencing adolescent’s vegetarian diet-related attitudes. Through self-administered loosely structured interviews, this research investigated factors potentially associated with vegetarian choices in adolescence. It checked (a) gender differences in vegetarian choices; (b) religious, familial, ethical, or health factors implied in vegetarian choices; and (c) indicators of well-being among young vegetarians. The findings suggest that for our sample, non-vegetarians have lower scores on health-related questions than others, while for vegetarian adolescents, the benefits of vegetarianism mainly depend on their ethical stances, beliefs, and values. Conversely, it is unrelated to factors such as the desire to lose weight, dissatisfaction about one’s body shape, or depressive feelings.

## 1. Introduction

Despite a growth trend of low meat consumers and vegetarians among the general population in Western countries [1,2,3], relatively few studies have investigated the prevalence and factors influencing vegetarian lifestyles among adolescents [4,5,6,7]. In Europe, vegetarianism is growing among adults, with estimates ranging from 2% in France to 9% in Germany and 12% in the UK [6,8]. However, data on children and adolescents are still scarce. Notably, in Italy, Ponzio et al. [9] estimate a 0.17% prevalence of vegetarians under the age of 18 (0.79% on average in the national sample) and report that the incidence reaches 0.96% in people over 65 (mainly females).

Baldassarre et al. [10], in a study conducted on 360 Italian families, found that 8.6% of mothers follow an alternative feeding regimen, such as different types of vegetarian diets or plant-based diets, and 9.2% of infants were weaned according to a vegetarian or vegan diet. Similarly, Talarico et al. [11] confirm that vegetarian children (1.01%) belong to families where parents adopt the same diet. In all cases, except for two, the children had vegetarian or vegan parents. A total of 2.88% of all parents of 1779 children were vegetarian and 1% were vegan. 

Vegetarianism is a more common dietary habit in women [12]. This prevalence is confirmed among adolescents: it is mainly girls who are vegetarians, who are more informed about vegetarianism, or who declare their interest in adopting a vegetarian diet [6,12,13].

Within the vegetarian population, there is a variety of dietary patterns. By definition, a vegetarian diet excludes all meat types, meat products (such as hams and sausages), fish, and seafood [14,15]. The consumption of eggs, dairy products, and honey is allowed (lacto-ovo-vegetarianism). In a narrower definition, strict vegetarians (or vegans) follow exclusively a plant-based diet and exclude wholly foods of animal origin (such as eggs, dairy products, and honey).

Conversely, according to a common definition, “semi-vegetarianism allows all kinds of foods (meat included) to be consumed, but with some limits of both quantity and frequency” [10]. In addition to the occasional consumption of meat, semi-vegetarians seem to differ from vegetarians in their beliefs: compared to vegetarians their food choices are rarely linked to issues such as animal welfare protection. Semi-vegetarianism among teenagers could be a trial or the first step towards a more stable and restricted vegetarian regimen [16]. Besides the heterogeneous eating patterns, vegetarians differ in the reasons underlying their choices.

Health reasons are the most common [17,18], but people choose vegetarianism for various philosophical, ecological, and religious beliefs [19]. Empirical studies evidence two main reasons for being or becoming a vegetarian: health or ethical motivations [13]. Health vegetarians focus on physical benefits such as a reduced risk of acquiring chronic diseases, living longer, or weight control [7,13].

Especially in contemporary Western culture, where meat predominates, with limited success in promoting non-meat diets, examining the motivation behind becoming a vegetarian is challenging, as it is a complex phenomenon. Ethical concerns, religious beliefs, and environmental factors are some of the factors that can guide the choice to change diet-related attitudes [12]. Health vegetarians gradually reduce meat consumption from diets but are generally less interested in switching to veganism. Ethical vegetarians are mainly supported by moral reasons, such as animal welfare, disgust associated with meat, and non-violence, or larger philosophical values such as equality and the respect for differences, or the idea that vegetarianism can contribute to reducing environmental pollution or world hunger [20,21,22,23].

Differences in beliefs and reasons for avoiding meat consumption have been observed between adult vegetarians and non-vegetarians, and health motives are the most common [18]. Less is known about adolescents’ reasons for choosing a vegetarian diet. In one of the first studies on this topic, Wright and Howcroft, in 1992 [24], found that among adolescents, emotional reasons associated with animal welfare, rather than health, are the basis of being vegetarian. 

Most studies investigating factors impacting adolescents’ food choices describe multiple factors that operate on multiple levels of influence [25]. Assuming Bronfenbrenner’s [26] ecological perspective, the factors can be described at different interacting levels: individual, including biological (e.g., weight) and psychological factors (e.g., attitudes, taste preferences, perceived barriers, etc.); interpersonal, such as family or peer influences (through modelling or perceived support); environmental (e.g., the availability and cost of foods, etc.); and cultural or macrosystem factors that play an indirect influence on food behaviours (e.g., mass media, religion, or societal norms of eating).

Vegetarian teenagers claim to know many vegetarian people [16], and often, the choice to be vegetarian is shared with parents and/or siblings [11].

Adolescence is a critical stage of physical growth and psychological development. Several studies have focused on the physical health of vegetarian adolescents and have generally showed benefits such as reduced overweight risks, type 2 diabetes’ onset, cardiovascular diseases, and some cancers [25,27,28,29,30].

A vegetarian lifestyle choice in adolescence, a diet rich in green fruits and vegetables, legumes, eggs, or low-fat milk, has long-term benefits on bones in adulthood [31]. A position statement of the American Dietetic Association and Dietitians of Canada [32] came to the conclusion that “appropriately planned vegetarian diets are healthful, nutritionally adequate, and provide health benefits in the prevention and treatment of certain diseases”. So, a balanced vegan diet can provide the nutritional requirements of preschool children [33]. However, there are also concerns about potential nutritional deficiencies in vegetarian diets, such as in all not well-balanced diets consumed in childhood, that, in the long run, could result in a restrictive regimen with adverse outcomes [31,34,35,36,37]. These concerns have led many scientific paediatric associations to spread recommendations for a well-planned and supplemented vegetarian/vegan diet [14,30,38,39,40].

Restrictive vegetarian diets have also been linked to eating disorders [37], since patients with eating disorders (mainly female) often report to have followed a form of a low-meat diet [41]. Similarly, ex-vegetarian adolescents are engaged in unhealthy weight control behaviours more often than adolescents who have never been vegetarians [42].

Abraham [43] observed that becoming a vegetarian may be a socially accepted way to lose weight. Especially in adolescence, attention to physical appearance and the search for slimness can lead teenagers to restrict their energy intake and adopt a meatless diet supported by the false belief that vegetarianism helps to lose weight [44,45,46].

However, the link between weight, body image satisfaction, and adolescent vegetarianism is unclear since, in some studies, vegetarians were more dissatisfied with their bodies [13], and in other studies, no differences emerged in weight preoccupation between vegetarian and non-vegetarian adolescents [45].

## 2. Aim of This Study

Although there is increasing research on various aspects of the physical health of vegetarian adolescents, including the risk of pathological eating practices [44], the adolescents’ perspective concerning vegetarian diets remains relatively unexamined. Therefore, the main scope of this study is to explore the profile (attitudes about nutrition and meat consumption, benefits and concerns for vegetarian diets, barriers or difficulties in changing personal lifestyles, and personal ethics) of adolescents who identified themselves as vegetarians or semi-vegetarians.

First, gender differences were explored to evaluate whether adolescents who define themselves as vegetarians (or semi-vegetarians), compared with their non-vegetarian peers, are more likely to be females (as observed in previous studies [6,12,13]). Secondly, studies with adults found differences in the quality of diet as a function of sexual orientation (i.e., more vegetable consumption among lesbian or bisexual women [47]. As far as we know, no studies have examined this factor in vegetarian adolescents. Therefore, the explorative question is whether sexual orientation could be associated with vegetarian choices among adolescents.

Third, we explored whether adolescents’ diet decisions are related to family or peer factors. Previous studies show that eating habits in the family play a critical role in following a vegetarian diet [11]. Therefore, we hypothesised that teenage vegetarians had relatives who were vegetarians as well. Similarly, due to the importance of peers’ support in adolescence [48], we hypothesised that adolescents who identified themselves as vegetarians or semi-vegetarians had more friends with the same food preferences than non-vegetarians.

Moreover, the recent literature suggests an association between vegetarianism/veganism and an increased risk of depression in adults [49]. Therefore, we also evaluated whether, as observed by Perry et al. [13], vegetarian adolescents report more frequent depressive feelings than non-vegetarian peers.

Finally, since a diet limiting meat consumption in adolescence might be considered a way of losing weight [41,42], a hypothesis to test is whether vegetarianism is associated with lower weight satisfaction or a high BMI.

## 3. Materials and Methods

### 3.1. Participants

This study involved a community sample of 1150 adolescents from public high schools in Catanzaro (the regional capital of Calabria, a southern region in Italy): 90.4% of the invited students returned the informed consent signed by their parents. Students came from upper secondary schools representing all Italian types of high schools: linguistic, classical, scientific, artistic, and social sciences high schools, technical institutes (with an economic and technological course of studies), and vocational schools (agricultural training). 

One section (i.e., five classes, from the first to the fifth year) from each school that agreed to collaborate on the study was chosen randomly. Therefore, the sampling procedure was convenient for the choice of schools and random as regards the choice of classes. The eligibility criteria for recruiting were belonging to one of the selected classes and receiving permission to participate from one’s parents. Some parents had difficulties filling in the questionnaires (i.e., not a native Italian speaker, a student with a certified intellectual disability, or learning disabilities). Twenty-one questionnaires, out of 1040, were excluded because they were incomplete or invalid. The sample was balanced by gender: 466 males (45.7%) and 553 females (54.3%). The range of age was 15 to 21 years, with a mean age of 17.8 (SD = 1.3) and 17.6 (SD = 0.9) for boys and girls, respectively.

### 3.2. Procedure

Informed consent forms were sent home to students’ parents (or guardians) with a brief presentation of the aim and procedure of the research. The students participated as volunteers by answering the self-report questionnaires anonymously. Participants authorised by the parents filled out the questionnaire during school hours (15–20 min). 

### 3.3. Measures

“Food choice, information and your attitudes” is a self-report questionnaire investigating several individual nutrition-related factors. The questionnaire was adapted to the Italian language from a previous study with adolescents [50] with the authors’ permission. For the purpose of the current study, the selected sections are the following (Appendix A).

“Your attitudes towards nutrition”. The original heading of this section of the questionnaire by Lea and Worsley [51] was “Your attitude towards nutrition and meat”. Here, this was simplified by omitting “meat” to introduce the measures selected for the topic of the current study. The first section (4 items) assesses attitudes towards nutrition, for example, “I frequently look for information on healthy eating”. The answers are on a 5-point Likert scale (1 = strongly disagree to 5 = strongly agree). This section explores the knowledge of diet in preventing illnesses and diseases or the attention related to the nutritional aspects of the type of food usually consumed, or again the ideas about healthy eating. A specific question asks, “Are you vegetarian now?”. Respondents can select one of three answers: (1) no, (2) yes, or (3) “I normally think of myself as being a semi-vegetarian”. This item was used to classify the students according to their self-identified dietary style and to submit the results for statistical analysis. The other two items explored family and peer factors: “Do you have any vegetarian family members? (None/father/mother/sibling/s/partner)” and “Do you have any vegetarian friends? (None/a few/about a quarter of my friends/about half of my friends/about three quarter of my friends/all)”.

Barriers to vegetarian diets. This section, with 21 items in the original questionnaire [51], investigates how difficult it could be to be a vegetarian (for example: “I would (or do) feel conspicuous among others” or “There is not enough iron in vegetarian diets”). The agreement is expressed on a 5-point Likert scale (from 1 = strongly disagree to 5 = strongly agree). In the Italian adaptation [50], the assessed issues are dietary difficulties (5 items, alpha = 0.76) that evaluate problems with a restricted food selection (e.g., “Vegetarian diets are not filling enough”); health-related concerns (5 items, alpha = 0.79) regarding nutritional deficiencies (proteins, calcium, etc.) resulting from plant-based foods (e.g., “There is not enough protein in vegetarian diets”); and family’s/friends’ food habits (2 items, alpha = 0.85), that is, “my friends/my family eat(s) meats”).

Benefits of vegetarian diets. A list of statements (24 items) evaluates the perceived benefits of a vegetarian diet (“I believe a vegetarian diet could (or does) help me to…”). Examples of items are “control my weight” or “create a more peaceful world”. The respondent rates his/her agreement on each item using a 5-point Likert scale (from 1 = strongly disagree to 5 = strongly agree). Three measurements are obtained [50]: health benefits (12 items, alpha = 0.92), that is, preventing diseases in general (e.g., heart disease and cancer) and promoting well-being (e.g., “increase my control over my health”); personal benefits (6 items, alpha = 0.83), both physical and psychological (“have plenty of energy” and “satisfying religious and/or spiritual needs”); and ethical benefits such as animal welfare, social equity, or environmental reasons (e.g., “the efficiency of food production”; 6 items, alpha = 0.84).

In our form of a reunified questionnaire, we inserted two items from the questionnaire “Health Behavior Questionnaire” [52,53] measuring depressive experiences and body satisfaction. Five items investigate the presence of depressive feelings, for example, “Over the past six months, how low have you felt about certain events?”. Responses varied on a 4-point Likert scale (1 = not agree to 4 = agree a lot). Satisfaction with weight was assessed by the question, “What do you think about your weight?”. The response options were the following: I would like to gain several kilos/to gain 2–3 kilos/my weight is right/to lose 2–3 kilos/to lose several kilos. 

A form for personal data and other information related to nutrition was inserted (age, gender, religion, and weight status). Among the personal data, we evaluated inserting one item asking about sexual orientation and height to obtain anthropometric statuses (Body Mass Index, BMI).

### 3.4. Statistical Analysis 

Based on self-reported dietary styles, the sample was divided into three groups, that is, non-vegetarians (NVs), vegetarians (Vs), and semi-vegetarians (SVs). The significance of differences in gender proportions among the three groups was tested by a contingency table 2 (male vs. female gender) × 3 (NV, V, and SV dietary style) and the chi-square test. Subsequently, a subgroup of participants was randomly selected from the broader non-vegetarian sample, matching them by age and gender proportion (16 males and 24 females) to the vegetarian and semi-vegetarian subgroups. Therefore, a series of crosstab chi-square tests tested the independence of dietary patterns with sociodemographic factors (i.e., gender, sexual orientation, and religion) and weight-related factors (i.e., BMI categories and weight satisfaction rates). Parametric variables according to dietary patterns were compared using F tests. 

Finally, group differences (NVs, Vs, and SVs) were estimated for all study measures by a one-way analysis of variance (ANOVA) for food attitudes and by a multivariate analysis of variance (MANOVA), or a non-parametric median test when variances were not homogeneous on Levene’s test, for the benefits/difficulties with a vegetarian diet.

All analyses were performed using IBM SPSS Statistics 22. For all tests, the *p* < 0.05 significance level was established.

## 4. Results 

### 4.1. Prevalence of the Vegetarian Diet and Gender Differences

Within the total sample of participants (n = 1019), 93.7% declared themselves non-vegetarian (n = 962), 43.9% male (n = 447) and 50.5% female (n = 515). Participants who defined themselves as semi-vegetarian (n = 39)—they occasionally eat foods of animal origin—were 3.9%, representing the choice of 1.6% males (n = 16) and 2.3% females (n = 23). Finally, 4.5% declared themselves vegetarian (n = 18), corresponding to 3% of males (n = 3) and 1.5% of females (n = 15) of the total sample. 

A chi-square test applied to a crosstab 2 (gender) × 3 (groups: NVs, Vs, and SVs) revealed the non-independence between factors [χ^2^ (2) = 6.68, *p* < 0.05].

### 4.2. Adolescents’ Characteristics According to Dietary Choices

Table 1 presents the differences among adolescents declaring different dietary choices on demographic, personal, health-related, and family/peer factors. The non-vegetarian (NV) adolescents (n = 40) were randomly selected from the wider group maintaining the proportion of males and females with the vegetarian (V) and semi-vegetarian (SV) participants. The three groups had similar gender distributions, ages, and sexual orientations. Mainly, regarding gender differences, even if this resulted in more girls than boys in the V group (24.2% and 8.6%, respectively), the crosstab chi-square test (2 gender × 3 groups) did not reach statistical significance. 

Regarding religion, this resulted in a significantly higher percentage of NVs who claim to be Christian. In contrast, adherence to non-traditional religions or none was more frequent among the Vs and SVs.

The proportion of vegetarian peers did not significantly differ among groups, but Vs and SVs were more likely to have a family member who follows the same diet. 

Finally, no statistically significant differences emerged as a function of diet for health indicators, that is, depressive feelings and weight-related factors (weight satisfaction and BMI).

### 4.3. Food Attitudes and Benefits/Difficulties with a Vegetarian Diet

Table 2 reports the descriptive statistics (M and SD) of participants’ responses to the sections of the questionnaire measuring attitudes, benefits, and difficulties with vegetarian diets [51]. 

Regarding attitudes, the ANOVAs revealed significant differences among the groups (NVs, Vs, and SVs) on two items: the importance of diet for prevention and the search for information on healthy eating. The post hoc multiple comparisons (Bonferroni’s tests for paired samples) indicate a lower accordance on diet for prevention by the NVs compared to the Vs [t(96) = −0.66, *p* = 0.04] and SVs [t(96) = −0.59, *p* = 0.02] and no differences between the Vs and SVs [t(96) = 0.07]. Similarly, the NVs reported lower scores for the need for information on healthy eating compared to the Vs [t(96) = −1.21, *p* = 0.004] and SVs [t(96) = −0.83, *p* = 0.01], but no differences resulted between the Vs and SVs [t(96) = 0.38]. 

The one-way MANOVA was conducted to test the differences in perceived benefits of a vegetarian diet among the groups. Statistically significant effects for the groups emerged on all the measures: health, personal, and ethical benefits. The post hoc multiple comparisons (Bonferroni’s tests for paired samples) indicated that the NVs reported lower scores on health benefits compared to the Vs [t(96) = −1.76, *p* < 0.001] and SVs [t(96) = −1.30, *p* < 0.001], and no differences were found between the Vs and SVs [t(96) = 0.46]. Secondly, the NVs reported lower scores for personal benefits compared to the Vs [t(96) = −1.24, *p* < 0.001] and SVs [t(96) = −0.66, *p* = 0.006], but no differences resulted between the Vs and SVs [t(96) = 0.58]. Finally, the NVs reported lower scores for ethical benefits compared to the Vs [t(96) = −1.90, *p* < 0.001] and SVs [t(96) = −1.02, *p* < 0.001]. Interestingly, vegetarian adolescents reported higher scores on ethical benefits than semi-vegetarians [t(96) = 0.89, *p* = 0.005].

Regarding measures on barriers, a series of non-parametric median tests was conducted because group distributions did not satisfy the homogeneity-of-variances assumption (Levene’s test). The median tests produced a refusal of the null hypothesis of the same medians for the three groups by dietary difficulties and health-related concerns, but they produced a null-hypothesis acceptance by family’s/friends’ habits.

As concerns dietary difficulties, the pairwise comparisons with Bonferroni’s error correction led to the rejection of the null hypothesis of the same distributions for the NVs vs. Vs and the NVs vs. SVs [χ^2^(1, N = 54) = 26.10, *p* < 0.001, and χ^2^(1, N = 79) = 53.52, *p* < 0.001, respectively], but not for the Vs vs. SVs [χ^2^(1, N = 57) = 4.5, *p* = 0.133].

Regarding health-related concerns, null-hypothesis rejections of the same distributions were observed in all pairwise comparisons with Bonferroni’s error correction: for the NVs vs. Vs [χ^2^(1, N = 54) = 19.08, *p* < 0.001], the NVs vs. SVs [χ^2^(1, N = 79) = 5.72, *p* < 0.05], and the Vs vs. SVs [χ^2^(1, N = 57) = 8.89, *p* < 0.01].

Finally, all pairwise comparisons (i.e., NVs vs. Vs, NVs vs. SVs, and Vs vs. SVs), with Bonferroni’s error correction applied to the family’s/friends’ habits measure, did not produce any significant results. 

## 5. Discussion

This original study contributes to understanding the individual factors underlying adolescents’ decision to be a vegetarian. It shows a comparable distribution and prevalence of vegetarian diet choices to those shown in a previous study conducted in the same Italian region [11]. However, the current study focuses on attitudes, beliefs, and ethical/health values linked to adolescents’ vegetarian choices. Furthermore, our non-clinical sample of enrolled Italian adolescents (n = 1019) allows for solid conclusions to be drawn.

In line with previous studies with the adult population, where vegetarianism is a more common dietary habit in women, adolescents who self-identify as vegetarian/semi-vegetarian are more likely to be females. Therefore, current data follow the growing interest in gender differences in dietary choices by enriching findings on the adolescent population [12]. 

No sexual orientation differences emerged between the non-vegetarian group and vegetarian/semi-vegetarian adolescents. These statistically non-significant results may be due to the small sample sizes of the groups. Moreover, as highlighted elsewhere [47], the link between diet patterns and sexual orientation is complex, and studies are still lacking. Therefore, further research is needed to consider identity development processes in adolescence.

Non-vegetarian adolescents show a more traditionalist religious creed, unlike vegetarians/semi-vegetarians who more frequently declare themselves atheists or are oriented towards other religions (such as pantheism). Adolescents’ food choices seem more linked to family members rather than friends. Vegetarian adolescents have siblings who follow the same diet, and semi-vegetarians have parents, siblings, or partners who are vegetarians. Therefore, despite the importance of peer relationships in adolescence, it does not seem that the choice of a vegetarian diet can be sustained by the need to imitate significant peers or by a simple adherence to the eating behaviours and values of friends. 

The results also reveal significant differences between non-vegetarians, semi-vegetarians, and vegetarians regarding their attitudes and health/ethical beliefs linked to the vegetarian lifestyle. 

Interesting findings emerge from adolescents’ responses to the questionnaire [50] assessing attitudes, barriers, and benefits associated with a vegetarian diet. The Attitudes section assessed beliefs relating to the general aspects of a healthy diet concerning nutritional characteristics, the prevention of diseases, and the diet’s influence on one’s health status. Non-vegetarian adolescents declared a low accordance about the importance of diet in preventing illnesses/diseases. They also have less knowledge about the benefits of healthy eating compared to semi-vegetarians and vegetarians. Interestingly, these two groups do not differ in giving importance to healthy eating and in the knowledge about nutrition’s power in preventing diseases. These findings confirm that vegetarian choices among adolescents are strongly sustained by adopting a healthy approach to diets.

This is also confirmed by the results of the sections Barriers and Benefits, in which youths provided feedback on how they perceive (“good” or “problematic”) the vegetarian diet approach. Notably, in the measure of benefits, they described their beliefs about health-positive aspects (e.g., the prevention of diseases such as heart disease and cancer); personal aspects (i.e., being more content with themselves and satisfying religious and/or spiritual needs); and ethical instances (such as a respect for the environment and the fight against inequalities, or animal welfare/rights). In the section Health, vegetarians and semi-vegetarians reported higher scores than non-vegetarians, and the adherence to health reasons did not differentiate teenagers who define themselves as strict vegetarians from semi-vegetarians. Adolescents who choose a vegetarian diet seem aware of the positive health consequences of excluding meat intake. Conversely, non-vegetarians share inadequate or superficial knowledge about the health benefits of vegetarianism, as observed in a previous study [54].

In the section Personal Benefits, vegetarians and semi-vegetarians provided similar answers, and both groups significantly differed from non-vegetarians who reported lower scores. These beliefs include a positive perception of vegetarian diets, such as food taste or energy value. In particular, the (expected) poor taste of plant-based foods, together with proximal factors (i.e., family support and diet), is one of the reasons declared by adolescents as evoking little interest in introducing more plant-based foods [55]. In the current study, the assessed personal benefits include psychological correlates such as a coherence with spiritual/religious values, self-esteem, and satisfaction with one’s choices. These perceived benefits and defining oneself as a “vegetarian” satisfy the psychological needs crucial for an adolescent’s identity and well-being. 

Finally, in the questions about ethical benefits, vegetarian adolescents reported higher scores than non-vegetarians and semi-vegetarians. These results suggest that a constellation of values oriented towards justice, nature, the promotion of a peaceful world, and respect for differences are most felt by vegetarian adolescents compared to semi-vegetarian adolescents. Alongside the already-known themes of animal rights and the refusal of any cause of suffering to other beings, an interesting ethical issue that emerges in the opinions of adolescents (mainly vegetarians), is the environmental impact and sustainability of vegetarian foods (e.g., less pollution, more efficient food production, or the fight against hunger). This aspect is little investigated among vegetarian adolescents [55], despite the growing sensitivity of young people towards environmental issues. Furthermore, findings from the current study have the advantage of increasing knowledge about the reasons underlying adolescents’ vegetarian lifestyles, but further studies are needed. 

The section Barriers assessed adolescents’ beliefs about why it may be challenging to be a vegetarian: dietary difficulties include motives such as a limited choice/energy from meat-free foods or the taste of meat; health-related concerns measure worries about nutritional deficiencies of the vegetarian diet, e.g., an insufficient balance of protein and/or micronutrients; and family’s/friends’ habits describe worries about diversity of dietary choices (mainly eating meat) among peers and/or family members.

Interestingly, no differences emerged among the groups according to their “environmental food style”, confirming that external influences (family or peers) do not seem to deter the potential decision to adopt a vegetarian lifestyle. Conversely, dietary and health-related difficulties differentiate adolescents according to their dietary patterns. Non-vegetarian adolescents report the most difficulties due to dietary restrictions and health-related issues. 

These findings suggest that an ethical direction could support being a “strict” vegetarian. Also, the presence of few concerns regarding the negative health consequences of meat-free eating is a chief suggestion. It demands educative paths to improve adolescents’ knowledge about health and nutrition needs. 

The semi-vegetarian adolescents are guided less by ethical reasons. However, we can assume that their more flexible approach to diets (the tolerance of the assumption of some animal foods) is driven by compensating for nutritional deficiencies or a too-restricted diet. These results extend, to the teenager populations, the previous findings on the adult population [56], according to which the semi-vegetarians differ from vegetarians in their belief system. Although health reasons are the most common trigger, semi-vegetarians less frequently attribute their food choices to issues related to animal welfare protection than strict vegetarians. An alternative explanation is that semi-vegetarianism may be a sort of testing or a first step towards a more stable restricted vegetarian food plan [16]. Further studies are needed with the scope to explore the complex connections between food-related knowledge, beliefs, and diet choices in adolescence [57].

Finally, in some studies, vegetarianism has been related to a socially accepted way of weight control [41,42]; this study compared body satisfaction and weight conditions in the function of adolescents’ diet choices. The results show no statistically significant differences among the non-vegetarians, semi-vegetarians, and vegetarians in the satisfaction with one’s weight and BMI. Similarly, the groups did not differ in the levels of depressive feelings reported by the participants. These results did not support the possible link between vegetarianism and low psychological well-being (e.g., anxiety and depression), thus confirming the adolescents’ findings with the adult community samples [58].

Some weaknesses of this observational study need to be underlined. In particular, the participants self-identified as vegetarians/semi-vegetarians, but this indicator may not correspond to the diet they adopt. Self-identified vegetarians (or vegans) sometimes or often violate their diet by consuming animal products [59]; therefore, as it is known, “being” vegetarian and eating vegetarian is not equivalent. However, the current study focuses on the adolescents’ perspectives towards diet choices, including one’s self-identity and personal beliefs, but not food intake. Therefore, although weak, self-presentation as a grouping criterion is consistent with the research aims.

Secondly, the interview overlooked a qualitative analysis of any associated food behaviour disorders, like eating disorders, such as anorexia and bulimia. Moreover, this study does not use specific indicators or more robust tools to assess the presence of other psychiatric disorders, such as depressive or anxiety disorders. 

However, this issue certainly needs long-term studies to improve the evaluation of the indicators of physical/psychological well-being, possibly investigating the quality of life and the consistency of decisions regarding the choice of a vegetarian diet among adolescents. It is also essential to consider that adolescents’ decision-making skills can change over time by age and environmental influences. Finally, it is important to point out that all participants came from a specified province in southern Italy. This convenience sample limits the study’s external validity and its power to generalise the results. Further studies with participants heterogeneous by geographical origin are then necessary, with the aim of checking whether other contextual factors (culture, local traditions, food availability, etc.) could influence vegetarian dietary styles in Italy. 

Despite these limitations, our study draws a picture of the current reasons supporting adolescents’ vegetarian choices. 

## 6. Implications

Understanding favourable factors/barriers to adopting a vegetarian diet or reducing meat consumption (as in the semi-vegetarians’ case) offers an essential insight for planning interventions. Health professionals (i.e., paediatricians and nutritionists), and other significant adults (such as teachers or sports trainers), should respect the dietary choices of adolescents and their families. This involves understanding the cultural, ethical, or religious motivations influencing everyone’s eating styles and not considering a plant-based diet as deviant or eccentric. In addition, health professionals should be guided by empirically based information about healthy vegetarian (or vegan) diets. Vegetarian adolescents generally know the health benefits of a meat-free diet. Still, they may fear not knowing how to manage it adequately, balancing the intake of nutrients contained in meat and derivatives. To promote a well-balanced vegetarian diet and prevent nutritional deficiencies, dietary-nutritional consultations could be offered to accompany adolescents in their decision-making process under the guidance of experts [38,40]. 

Community interventions (i.e., school-based ones) can effectively disseminate correct nutritional education. However, it is known that increasing knowledge (i.e., providing information on the benefits of a plant-based diet by nutritionists) is not an intervention that guarantees the adoption of healthy behaviours among adolescents [53]. Conversely, practical strategies for promoting a healthy diet could be implemented in educational programs. A study [60] explored factors affecting the preference for vegetables among high school students. Regardless of the adolescents’ weight status (BMI), the personal factors determinant for increasing vegetable intake resulted in positive attitudes towards plant-based foods and self-efficacy, that is, adolescents’ confidence to make their own choices and achieve personal goals. Among the environmental factors, vegetable accessibility and positive experiences with plant-based foods resulted in determinants for adolescents’ choices. These findings suggest the importance of supporting adolescents’ positive attitudes and experiences with plant-based foods and improving personal skills (such as self-efficacy and decision making). Interestingly, learning to cook vegetarian foods and discovering that they are appetising can also increase adolescents’ life skills (e.g., self-efficacy, perception of control over health, etc.) and psychological well-being [55].

Similarly, active interventions (such as focus-group ones [54]) can effectively offer participants opportunities to openly express their beliefs about healthy nutrition or to share positive experiences that can motivate a shift in dietary patterns [61]. Education might also modify adolescents’ barriers or misconceptions regarding a plant-based diet, for example, parental influences or the expectation of the tastiness of vegetable foods compared with a good flavour of the meat [55].

Finally, the practical implications of the current study derive from the weight attributed by vegetarian adolescents (less by semi-vegetarians) to ethical motivations, particularly the sustainability and ecological benefits of vegetarian/vegan diets. The issues of the low environmental impact of plant-based foods (against the damage of intensive animal breeding for pollution and climate change) are becoming popular among young people [18]. Values associated with a vegetarian lifestyle are social equity and the accessibility of alternatives to meat among most people worldwide against hunger and inequality. Future research based on community qualitative (or mixed) methods could explore these emerging ethical issues, with the scope to understand their weight in driving adolescents’ dietary decisions and guide policies and interventions. For example, we suggest that future researchers use Online Photovoice (OPV) [62,63] to research the same or similar topics. It is an innovative qualitative research method that allows for the participants to express their own experiences with as little manipulation as possible, compared to traditional quantitative methods. 

## 7. Conclusions

The current study explored how vegetarianism among adolescents aligns with various health, ethical, and environmental sustainability values. The findings indicate that ethical considerations primarily motivate adolescents who self-identify as “strict” vegetarians (compared with semi-vegetarians), while the perceived health benefits are considered additional advantages. Among individual factors associated with vegetarianism, a predominance of girls and a low adherence to traditional religions resulted. Vegetarian adolescents more often come from families where others follow meat-free diets. Regarding well-being indicators, vegetarianism is not associated with adolescents’ desire for weight loss or body dissatisfaction.

Gaining autonomy and making choices consistent with one’s goals and values are critical developmental tasks in adolescence. Since being self-determined increases personal well-being and can lead to better neuropsychological developments [64,65], it is essential to support adolescents’ dietary choices and provide information about a balanced diet, the correct vitamin reintegration, and dietary protein intake, as the scientific community recommends. 

Interventions should prevent health and nutritional problems associated with unbalanced dietary choices but also support adolescents’ dietary lifestyles as a component of a more complex view of the world oriented towards ethical values, animal welfare, the fight against inequalities, and a defence of the environment. 

## Figures and Tables

**Table 1 healthcare-11-02885-t001:** Characteristics of adolescents according to dietary choices.

Variables		Non-Vegetarians	Vegetarians	Semi-Vegetarians	Comparisons [χ^2^ or F]
		n = 40 (41.2%)	n = 18 (18.6%)	n = 39 (40.2%)	
Gender, n (%)	Boys	16 (45.7%)	3 (8.6%)	16 (45.7%)	χ^2^ (2) = 3.62 *p* = 0.64
	Girls	24 (38.7%)	15 (24.2%)	23 (37.1%)	
Age (years), Mean ± SD		17.73 ± 0.11	17.89 ± 0.02	17.62 ± 0.10	F (2, 94) = 0.40 *p* = 0.67
Religion, n (%)	Christian Muslim Buddhist Jewish Agnostic/Atheist Other	34 (61.8%) 0 (0.0%) 0 (0.0%) 0 (0.0%) 6 (16.7%) 0 (0.0%)	4 (7.3%) 0 (0.0%) 0 (0.0%) 0 (0.0%) 12 (33.3%) 2 (50%)	17 (30.9%) 0 (0.0%) 1 (100%) 1 (100%) 18 (50%) 2 (50%)	χ^2^ (8) = 27.82 *p* < 0.001 9 (60%) expected count cells <5
Sexual orientation, n (%)	Homosexual Bisexual Heterosexual Asexual	1 (10.0%) 2 (20.0%) 36 (48.0%) 1 (50.0%)	4 (40.0%) 3 (30.0%) 11 (14.7%) 0 (0.0%)	5 (50.0%) 5 (50.0%) 28 (37.3%) 1 (50.0%)	χ^2^ (6) = 9.17 *p* = 0.16 9 (75%) expected count cells <5
Vegetarians in family, n (%)	None Father Mother Sibling/s Boy/girlfriend	40 (46.0%) 0 (0.0%) 0 (0.0%) 0 (0.0%) 0 (0.0%)	13 (14.9%) 0 (0.0%) 0 (0.0%) 5 (83.3%) 0 (0.0%)	34 (39.1%) 1 (100%) 2 (100%) 1 (16.7%) 1 (100%)	χ^2^ (8) = 23.93 *p* < 0.02 12 (80%) expected count cells <5
Vegetarian peers, n (%)	No, none Yes	23 (47.9%) 17 (34.7%)	5 (10.4%) 13 (26.5%)	20 (41.7%) 19 (38.8%)	χ^2^ (2) = 4.47 *p* = 0.11
Weight satisfaction, n (%) [*I would like…..*]	to gain several kilos to gain 2–3 kilos my weight is right to lose 2–3 kilos to lose several kilos	0 (0%) 4 (44.4%) 14 (51.9%) 13 (32.5%) 8 (44.4%)	0 (0%) 1(11.1%) 5 (18.5%) 6 (15.0%) 6 (33.3%)	2 (100%) 4 (44.4%) 8 (29.6%) 21 (52.5%) 4 (22.2%)	χ^2^ (8) = 10.86 *p* = 0.21 7 (46.7%) expected count cells <5
Depressive feelings, Mean ± SD		11.58 ± 3.52	12.00 ± 3.97	12.92 ± 3.72	F (2, 94) = 1.35 *p* = 0.26
BMI (kg/m^2^) ^1^, n (%)	Underweight Normal weight Overweight Obese	7 (50.0%) 25 (35.2%) 7 (77.8%) 1 (50.0%)	3 (0.3%) 13 (1.3%) 1 (0.1%) 1 (0.1%)	4 (28.6%) 33 (46.5%) 1 (11.1%) 0 (0.0%)	χ^2^ (6) = 9.10 *p* = 0.17 7 (58.3%) expected count cells <5

Note: ^1^ BMI criteria (underweight: BMI < 18.49; normal: 18.50 ≥ BMI ≤ 24.99; overweight: 25.00 ≥ BMI ≤ 29.99; obese: BMI ≥ 30.00).

**Table 2 healthcare-11-02885-t002:** Attitudes, perceived benefits, and barriers to a vegetarian diet (M ± SD). Comparisons among the groups.

Measures		Non-Vegetarians (n = 40)	Vegetarians (n = 18)	Semi-Vegetarians (n = 39)	Comparisons [F or χ^2^]
Attitudes	Diet is important in preventing illness and disease	3.90 ± 1.06	4.56 ± 0.78	4.59 ± 0.79	F(2, 96) = 5.30, *p* < 0.01
	I do not need to make any changes to the food I eat as it is already healthy enough	2.75 ± 1.32	2.72 ± 1.27	2.67 ± 1.29	n.s.
	I usually do not think about the nutritional aspects of the types of food that I eat	2.63 ± 1.33	1.78 ± 1.11	2.36 ± 1.51	n.s.
	I frequently look for information on healthy eating	2.40 ± 1.24	3.61 ± 1.30	3.23 ± 1.31	F(2, 96) = 7.09, *p* < 0.001
Benefits	Health benefits	2.58 ± 0.82	4.34 ± 0.67	3.88 ± 0.77	F(2, 94) = 43.30, *p* < 0.001 Eta_p_^2^ = 0.48
	Personal benefits	1.94 ± 0.91	3.18 ± 1.08	2.60 ± 0.81	F(2, 94) = 12.59, *p* < 0.001 Eta_p_^2^ = 0.21
	Ethical benefits	2.33 ± 1.09	4.23 ± 0.88	3.35 ± 0.84	F(2, 94) = 26.78, *p* < 0.001 Eta_p_^2^ = 0.36
Barriers	Dietary difficulties	3.16 ± 0.78	1.21 ± 0.39	1.57 ± 0.55	χ^2^(2, N = 97) = 51.09 *p* < 0.001
	Health-related Concerns	2.61 ± 1.02	1.22 ± 0.31	1.87 ± 0.78	χ^2^(2, N = 97) = 20.21, *p* < 0.001
	Family’s/friends’ habits	2.95 ± 1.51	1.97 ± 1.43	2.70 ± 1.34	χ^2^(2, N = 97) = 3.29 n.s.

## Data Availability

The dataset analysed in this study can be requested from the corresponding author on reasonable request.

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
