# Peer review of "Diet-Related Attitudes, Beliefs, and Well-Being in Adolescents with a Vegetarian Lifestyle"

_healthcare, 2023, doi:10.3390/healthcare11212885_

Round 1
Reviewer 1 Report
Comments and Suggestions for Authors
The study identifies young people’s diet-related attitudes and behaviour in the Southern Italian Region.
The topic is highly relevant for various fields, including public health.
However, there are several issues I would like to raise regarding the paper.
1.
Firstly, the paper needs careful language and style editing, as there are numerous examples of missing full stops, capitals without a need (even in the title) and inconsistent capitalization of identical terms.
A few examples are shown in the points below.
2.
Title: “Teenagers' beliefs and well-being in dietary decisions motivate them to choose a Vegetarian lifestyle.”
One issue is with the title. First, there are not only teenagers in the sample. Second, a full stop is not needed. Third, the capitalization of vegetarian is inconsistent (I suggest not using capital letters). Fourth, the title is vague about what the study examined (also see point below). Fifth, the region should be added to the title.
3.
Abstract:
“Despite the spread of vegetarianism in the Western world, researchers have only investigated the benefits and risks of physical health.”
That is not true, as many studies on psychological outcomes related to non-meat diets exist. I suggest examining the literature, citing needed references in the text, and changing the wording in the abstract.
“But very few studies are available on the factors influencing this choice among adolescents.”
Which choice?
“The findings suggest that for many vegetarian adolescents, the perceived health advantages represent additional benefits of vegetarianism.”
Additional over what? Please state the percentage, as “many” is too vague.
“However, vegetarianism mainly depends on their ethical stances, beliefs, and values.”
Whose?
4.
“Baldassarre et al. [10] found that 8.6% of Italian mothers declare an alternative regimen,”
Such as?
5.
“Similarly, Talarico et al. [11] confirm that vegetarian children (1.01%) belong to families where parents adopt the same diet.
Only 1.01% of vegetarian children have vegetarian parents?
6.
“Vegetarianism is a predominantly female phenomenon [12].”
A female phenomenon.
7.
“In a narrower definition, strict vegetarians (or vegans) follow exclusively a plant-based diet and exclude wholly foods of animal origin (including eggs, dairy products, and honey).”
“Including” is misleading here, as the prior word is excluding. Change to “such as”.
8.
“Empirical studies evidence two reasons for being or becoming a vegetarian: health or ethical motivations [13].”
Two main reasons.
9.
“The ideology supporting their dietary choice is the maintenance of traditions, conformism, and societal values,”
Unclear. Vegetarian diets and associated values are considered non-traditional, alternative and non-conformist in Western societies.
10.
“… adult vegetarians and non-vegetarians, and health is generally the most significant boost [18]”
Boost to what and to whom?
11.
“A vegetarian lifestyle choice in adolescence, a diet rich in green fruits and vegetables, legumes, eggs, or low-fat milk,…”
Lacto-ovo vegetarian. Please also add a reference on vegan children’s outcomes (which tend to be, in the case of balanced vegan diets, positive).
12.
“However, there are also concerns about potential nutritional deficiencies that, in the long run, could result from a restrictive or unbalanced vegetarian diet starting in childhood [30]–[34].”
Yes, but please state that the need for balanced diets goes for all diets, including meat diets, and is not particular for only vegan/vegetarian diets.
13.
“Restrictive vegetarian diets have also been linked to eating disorders [34], since patients with eating disorders (mainly female) often report to have followed a form of low-meat diet [38].”
Yes, but please state the causality. People with eating disorders more often turn to vegan/vegetarian diets (than meat-eaters) than vegans/vegetarians become ill people with eating disorders.
14.
“Especially in adolescence, attention to physical appearance and the search for slimness can lead teenagers to restrict their energy intake and adopt a meatless diet supported by the false belief that vegetarianism helps to lose weight [41][42].”
There are several studies indicating that vegan (and plant-based) diets are associated with lower BMI (see M. Greger’s “How not to diet” for scientific references).
15.
Aim the study
This is the more problematic part of the paper. The aims are very broadly written, and many of the aims covered (e.g., link with sexuality, etc.) are not covered in the Introduction/Theoretical part. Each of the predictors examined should have a short literature review of previous findings.
The aims should be specific.
“…adolescents who identified themselves as vegetarians or semi-vegetarians”.
These should be written in the title. The self-identity studies show that self-identified vegans/vegetarians often consume animal products, which makes self-identity indicator a relatively poor indicator of actual diet. This should also be stated in the study limitations section in the Discussion.
16.
Sample description.
Authors should state the following: how was a sample size calculated (was the sample representative? If so, of what populatin=); which sampling method was used. Explain how the study size was arrived at. Give the eligibility criteria and the sources and methods of selection of participants. Of all invited students, how many did fill out the questionnaire, etc.
17.
“Of the total collected questionnaires, 21 were excluded because they were incomplete or invalid.”
Was any analysis performed regarding whether/how removing the incomplete etc., cases impacted the findings?
18.
The authors may take advantage of MDPI journals and other publications on determinants of dietary patterns and include them for an interested reader, contextualizing the results by comparing them with studies, especially those regarding the role of education (authors themselves state that educational interventions are needed: “It demands educative paths to improve adolescents' knowledge about health and nutrition needs.”). See, for example:
1) https://doi.org/10.3390/su132313036
2) Gossard, M.H.; York, R. Social Structural Influences on Meat Consumption.Hum. Ecol. Rev.2003,10, 1–9.
3) Koch, F.; Heuer, T.; Krems, C.; Claupein, E. Meat consumers and non-meat consumers in Germany: A characterization based on rresults of the German National Nutrition Survey II.J. Nutr. Sci.2019,8, e21
In addition, where there differences between school programs with regard to self-identity and other outcomes?
19.
Many of the findings (group differences) are not significant. Yet that may be due to the small sample sizes of the three groups? Please elaborate to the reader.
20.
“Your attitudes towards nutrition:”
Delete Yours. Always use small (non-capital) letters when not starting a sentence.
21.
Add all the items in the supplementary material for the interested reader.
22.
“(from 1= Strongly disagree to 5 = strongly agree”
Correct capitalization and inconsistent spacing (in the whole paper).
23.
““Over the past six months, how low have you felt about certain events?”. Response varies on a 4-point Likert scale (from 1= = not agree, to 4=agree a lot).”
What were the depression items? Please, put in the supplementary material.
25.
A reader would welcome more concrete policy recommendations stemming from the study findings.
26.
“Differently from the adult population, where vegetarianism is a predominantly female phenomenon [12], there were no significant differences in gender between the non-vegetarian group and vegetarian/semi-vegetarian adolescents.”
Readers would welcome a discussion on potential reasons for the lack of gender differences.
28.
“These findings confirm that the vegetarian choice among adolescents is strongly sustained by adopting a healthy approach to life.”
Vague, unclear.
29.
“This is also confirmed by the results of the sections Barriers and Benefits, in which youths provided feedback on how they perceive (“good” or “problematic”) the vegetarian diet approach. Particularly, in the measure of benefits, they described their beliefs about health-positive aspects (such as the prevention of disease, e.g. heart disease, cancer), personal aspects (i.e. being more content with themself, satisfying religious and/or spiritual needs), and ethic instances (such as the respects of the environment, the fight against inequalities, or animal welfare/rights). In the section Health (alpha= .92 in the current study), vegetarians and semi-vegetarians reported higher scores than non-vegetarians. Similarly, in the section Personal benefits (alpha=. 83), vegetarians and semi-vegetarians significantly differ from non-vegetarians who reported lower scores. Finally, in the questions about Ethical benefits (alpha= .84),”
Alphas do not belong in the Discussion sections but in the Method section. Second, capitalization. Third, clearly state what construct was measured, not stating vague names of the section. Fourth, summarize key results, and be concrete. Goes for the next paragraphs as well.
31.
“Non-vegetarian adolescents utter the greatest difficulties due to dietary restrictions and health-related issues.”
Change “utter” to “report”.
32.
“authors firmly believe”
Please, delete this phrasing.
33.
“Institutional Review Board Statement: The study was conducted following the Declaration of Helsinki, and the ethical standard norms for psychological research of the Italian Psychology Association (AIP; chrome-extension://efaidnbmnnnibpcajpcglclefindmkaj/https://aipass.org/wp-content/uploads/2023/02/Codice-Etico_luglio-2022.pdf).”
The link states Chrome extensions? Please, correct.
34.
“Data Availability Statement: The data presented in this study are available in the section 3. Materials and Methods, and 4. Results in the article.”
It should be stated that survey data is available from the first (or lead) author upon reasonable request (please, see other MDPI papers).
35.
““C. Larsson, “Young vegetarians and omnivores Dietary habits and other health-related aspects”.”
Please, provide all the needed information of all references in the references list.
36.
Considering the above-stated comments, I suggest the authors revise the paper.
Comments on the Quality of English LanguagePlease, see above.
Author Response
Reviewer 1
The study identifies young people’s diet-related attitudes and behaviour in the Southern Italian Region.
The topic is highly relevant for various fields, including public health.
However, there are several issues I would like to raise regarding the paper.
- Firstly, the paper needs careful language and style editing, as there are numerous examples of missing full stops, capitals without a need (even in the title) and inconsistent capitalization of identical terms.
A few examples are shown in the points below.
2.Title: “Teenagers' beliefs and well-being in dietary decisions motivate them to choose a Vegetarian lifestyle.”
One issue is with the title. First, there are not only teenagers in the sample. Second, a full stop is not needed. Third, the capitalization of vegetarian is inconsistent (I suggest not using capital letters). Fourth, the title is vague about the study's examination (also see point below). Fifth, the region should be added to the title.
1-2 Answer: Thanks for comments, but we are not totally agreeing with them.
We have changed the title on the basis of your comments; we also replaced “teenagers” with “adolescents”, so the upper age limit is more nuanced, and removed the full stop and capitalization. So we changed the title in: “Diet-related attitudes, beliefs and well-being in adolescents with a vegetarian lifestyle”
We do not agree with introducing the regional reference of the participants in the title, because this reference is unusual in literature unless one wants to accentuate the local specificity of a study. This is not our case.
Arnett (2008) reminded the scientific community that 67% of all psychological research in the US was conducted with university students: this did not mean that the authors are asked to always specify in the title if participants were university students. Similarly, if we examine the literature on vegetarianism/veganism, study titles do not indicate the geographical location of the participants.
However, since participants come from a limited geographical area, a new comment on this characteristic is discussed within the study's limits.
It is also essential to consider that adolescents’ decision-making skills can change over time, by age and environmental influences. Finally, it is important to point out that all participants came from a specified province in southern Italy. This convenience sample limits the study's external validity and its power to generalise the results. Further studies with participants heterogeneous by geographical origin are then necessary, with the aim of checking whether other contextual factors (culture, local traditions, food availability, etc.) could influence vegetarian dietary style in Italy. Despite these limitations, our study drew a picture of the current reasons supporting adolescents’ vegetarian choices.
3.Abstract: “Despite the spread of vegetarianism in the Western world, researchers have only investigated the benefits and risks of physical health.”
That is not true, as many studies on psychological outcomes related to non-meat diets exist. I suggest examining the literature, citing needed references in the text, and changing the wording in the abstract.
“But very few studies are available on the factors influencing this choice among adolescents.”
Which choice?
“The findings suggest that for many vegetarian adolescents, the perceived health advantages represent additional benefits of vegetarianism.”
Additional over what? Please state the percentage, as “many” is too vague.
“However, vegetarianism mainly depends on their ethical stances, beliefs, and values.”
Whose?
3 Answer: Thank you for the corrections. In agreement with what you proposed, we have modified some sentences as follows: “Considering the spread of non-meat diets in the Western world, researchers have investigated ​​the benefits and risks to physical and psychological health. Despite this, few studies have been conducted on factors influencing adolescent’s vegetarian diet-related attitudes.”
Moreover, we agree that "many" is too vague. So, about what was requested, we wanted to specify that the questionnaire administered does not allow us to derive a percentage. As a Likert scale, it allows us only to estimate the strength of adherence to a statement. We have more faithfully reported the results of our study in this way: “The findings suggest that for our sample, non-vegetarians have lower scores on health-related questions than others, while for vegetarian adolescents, the benefits of vegetarianism mainly depend on their ethical stances, beliefs, and values. Conversely, it is unrelated to factors such as the desire to lose weight, dissatisfaction about body shape or depressive feelings.”
4.“Baldassarre et al. [10] found that 8.6% of Italian mothers declare an alternative regimen,”
Such as?
Answer 4. Thank you for pointing out the incompleteness of the information. The text has been reworded to specify which alternative regimes were referred to. We modified so “Baldassarre et al. [10], in a study conducted on 360 Italian families, found that 8.6% of mothers follow an alternative feeding regimen such as different types of vegetarian diets or plant-based diets, and 9.2% of infants were weaned according to a vegetarian or vegan diet.”
- “Similarly, Talarico et al. [11] confirm that vegetarian children (1.01%) belong to families where parents adopt the same diet.
Only 1.01% of vegetarian children have vegetarian parents?
Answer 5. No, 2.88% of all parents of 1779 children were vegetarian and 1% vegan.
In all cases of vegetarian children, except for two, the children had vegetarian or vegan parents. We have reshaped the data in the text.
- “Vegetarianism is a predominantly female phenomenon [12].” A female phenomenon.
Answer 6. Thanks. We correct in ”is a more common dietary habit in women”
7.“In a narrower definition, strict vegetarians (or vegans) follow exclusively a plant-based diet and exclude wholly foods of animal origin (including eggs, dairy products, and honey).”
“Including” is misleading here, as the prior word is excluding. Change to “such as”.
Answer 7. Thanks for the correction, as suggested we changed the term.
8.“Empirical studies evidence two reasons for being or becoming a vegetarian: health or ethical motivations [13].” Two main reasons.
Answer 8. Thanks for the suggestion. We have adjusted the phrase.
9.“The ideology supporting their dietary choice is the maintenance of traditions, conformism, and societal values,”
Unclear. Vegetarian diets and associated values are considered non-traditional, alternative and non-conformist in Western societies.
Answer 9. Thanks for the note. There is probably a linguistic inconsistency. Instead, we intended a greater mind openness to ethical and cultural social values for vegetarian adolescents than non-vegetarians. Following the cited article, we have modulated the sentence below: …”Especially in contemporary Western culture, where meat predominates, with limited success in promoting non-meat diets, examining the motivation behind becoming a vegetarian is challenging, as it is a complex phenomenon. Ethical concerns, religious beliefs, and environmental factors are some of the factors that can guide the choice to change diet-related attitudes”
10.“… adult vegetarians and non-vegetarians, and health is generally the most significant boost [18]”
Boost to what and to whom?
Answer 10. Thanks, we changed to “health motives are the most common”
11.“A vegetarian lifestyle choice in adolescence, a diet rich in green fruits and vegetables, legumes, eggs, or low-fat milk,…”
Lacto-ovo vegetarian. Please also add a reference on vegan children’s outcomes (which tend to be, in the case of balanced vegan diets, positive).
Answer 11. Thanks, we have added the necessary references:
A position statement of the American Dietetic Association and Dietitians of Canada [31] came to conclusion that ‘appropriately planned vegetarian diets are healthful, nutritionally adequate, and provide health benefits in the prevention and treatment of certain diseases’. So, a balanced vegan diet can provide the nutritional requirements of preschool children. [32]
[32] “Position of the American Dietetic Association and Dietitians of Canada: Vegetarian diets,” J Am Diet Assoc, vol. 103, no. 6, pp. 748–765, Jun. 2003, doi: 10.1053/jada.2003.50142.
[33] M. E. Kiely, “Risks and benefits of vegan and vegetarian diets in children,” Proceedings of the Nutrition Society, vol. 80, no. 2, pp. 159–164, May 2021, doi: 10.1017/S002966512100001X.
- “However, there are also concerns about potential nutritional deficiencies that, in the long run, could result from a restrictive or unbalanced vegetarian diet starting in childhood [30]–[34].”
Yes, but please state that the need for balanced diets goes for all diets, including meat diets, and is not particular for only vegan/vegetarian diets.
Answer 12. Thanks, we modified in.. “However, there are also concerns about potential nutritional deficiencies in vegetarian diets, such as in all not well-balanced diets consumed in childhood, that, in the long run, could result in a restrictive regimen with adverse outcomes”
- “Restrictive vegetarian diets have also been linked to eating disorders [34], since patients with eating disorders (mainly female) often report to have followed a form of low-meat diet [38].”
Yes, but please state the causality. People with eating disorders more often turn to vegan/vegetarian diets (than meat-eaters) than vegans/vegetarians become ill people with eating disorders.
Answer 13. Thank you for the clarification. The causal link would be interesting to investigate, but it is not the deepening here. It is complex to go into, and we have chosen not to deal with it in this study.
- “Especially in adolescence, attention to physical appearance and the search for slimness can lead teenagers to restrict their energy intake and adopt a meatless diet supported by the false belief that vegetarianism helps to lose weight [41][42].”
There are several studies indicating that vegan (and plant-based) diets are associated with lower BMI (see M. Greger’s “How not to diet” for scientific references).
Answer 14. Thank you, we are aware. We corrected the answer in this way: “Especially in adolescence, attention to physical appearance and the search for slimness can lead teenagers to restrict their energy intake and adopt a meatless diet supported by the false belief that vegetarianism helps to lose weight [44][45].”
15.
Aim the study
This is the more problematic part of the paper. The aims are very broadly written, and many of the aims covered (e.g., link with sexuality, etc.) are not covered in the Introduction/Theoretical part. Each of the predictors examined should have a short literature review of previous findings.
The aims should be specific.
Answer 15. Thank you for the suggestions, we have re-written the section as follows:
First, gender differences were explored to evaluate whether adolescents who define themselves as vegetarians (or semi-vegetarians), compared with their nonvegetarian peers, are more likely to be females (as observed in previous studies [6], [12], [13]. Secondly, studies with adults found differences in the quality of diet as a function of sexual orientation (i.e., more vegetable consumption among lesbian or bisexual women [48]). As far as we know, no studies have examined this factor in vegetarian adolescents. Therefore, the explorative question is whether sexual orientation could be associated with vegetarian choice among adolescents.
Third, we explored whether adolescents’ diet decisions related to family or peer factors. Previous studies show that eating habits in the family play a critical role in following a vegetarian diet [11] Therefore, we hypothesised that teenage vegetarians had relatives who were vegetarians as well. Similarly, due to the importance of peers’ support in adolescence [49], we hypothesised that adolescents who identified themselves as vegetarians or semi-vegetarians had more friends with the same food preference than non-vegetarians.
Moreover, recent literature suggests an association between vegetarianism/veganism and an increased risk of depression in adults [50]. Therefore, we also evaluated whether, as observed by Perry et al. [13], vegetarian adolescents report more frequent depressive feelings than non-vegetarian peers.
Finally, since a diet limiting meat consumption in adolescence might be considered a way of losing weight [41] [42], the hypothesis to test is whether vegetarianism is associated with lower weight satisfaction or a high BMI.
15 – self-identified vegetarians
“…adolescents who identified themselves as vegetarians or semi-vegetarians”.
These should be written in the title. The self-identity studies show that self-identified vegans/vegetarians often consume animal products, which makes self-identity indicator a relatively poor indicator of actual diet. This should also be stated in the study limitations section in the Discussion.
Answer: Thank you for this observation. A comment was added among study limitations.
“Some weaknesses of this observational study need to be underlined. In particular, participants self-identified as vegetarians/semi-vegetarians, but this indicator may not correspond to the diet they adopt. Self-identified vegetarians (or vegans) sometimes or often violate their diet by consuming animal products [60]; therefore, as it is known, “being” vegetarian and eating vegetarian is not equivalent. However, the current study focuses on the adolescents’ perspectives towards diet choice, including self-identity and personal beliefs, but not food intake. Therefore, although weak, self-presentation as a grouping criterion is consistent with the research aims.”
16.
Sample description.
Authors should state the following: how was a sample size calculated (was the sample representative? If so, of what populatin=); which sampling method was used. Explain how the study size was arrived at. Give the eligibility criteria and the sources and methods of selection of participants. Of all invited students, how many did fill out the questionnaire, etc.
Answer 16: Thanks for the clarifying suggestions.
We modified the Participants description as follows:
“The study involved a community sample of 1150 adolescents from public high schools in Catanzaro (the regional capital of Calabria, a southern region in Italy): 90.4% of the invited students returned the informed consent signed by their parents. Students came from upper secondary schools representing all Italian types of high schools: linguistic, classical, scientific, artistic, and social sciences high schools, technical institutes (with economic and technological course of studies), and vocational schools (agricultural training).
One section (i.e., five classes, from the first to the fifth year) from each school that agreed to collaborate on the study was chosen randomly. Therefore, the sampling procedure was convenient for the choice of schools and random as regards the choice of classes. The eligibility criteria for recruiting were belonging to one of the selected classes and receiving permission to participate from parents. Some parents had difficulties filling in the questionnaires (i.e., not a native Italian speaker, a student with a certified intellectual disability or learning disabilities). Twenty-one questionnaires out of 1040, were excluded because they were incomplete or invalid. The sample was balanced by gender, 466 males (45.7%) and 553 females (54.3%). The range of age was 15 to 21 years, with a mean age of 17.8 (SD=1.3) and 17.6 (SD= .9) for boys and girls, respectively.”
17.
“Of the total collected questionnaires, 21 were excluded because they were incomplete or invalid.”
Was any analysis performed regarding whether/how removing the incomplete etc., cases impacted the findings?
Answer 17: No, it was not, because 21 questionnaires are only the 2% from total and the study was exploratory: we were interested to the actual answers by respondents.
- The authors may take advantage of MDPI journals and other publications on determinants of dietary patterns and include them for an interested reader, contextualizing the results by comparing them with studies, especially those regarding the role of education (authors themselves state that educational interventions are needed: “It demands educative paths to improve adolescents' knowledge about health and nutrition needs.”). See, for example:
1) https://doi.org/10.3390/su132313036
2) Gossard, M.H.; York, R. Social Structural Influences on Meat Consumption.Hum. Ecol. Rev.2003,10, 1–9.
3) Koch, F.; Heuer, T.; Krems, C.; Claupein, E. Meat consumers and non-meat consumers in Germany: A characterization based on rresults of the German National Nutrition Survey II.J. Nutr. Sci.2019,8, e21
In addition, where there differences between school programs with regard to self-identity and other outcomes?
Answer 18. Thanks for the keen observation. In our sample, we did not investigate the difference in school programs in general, nor whether regarding self-identity. With pleasure, we searched and read the articles you suggested. We have noted that there are interesting correlations between the choice not to eat meat and the years of education or the age of the population examined.
Unfortunately, these aspects were not taken into account in our study because we did not have access to the educational programmes, although we like the idea of being able to deepen later.
- Many of the findings (group differences) are not significant. Yet that may be due to the small sample sizes of the three groups? Please elaborate to the reader.
Answer 19. Thank you for the question, actually, there are several significant differences already, see the results and discussion section. In addition, we would like to point out the added comment regarding the association with sexual orientation in the discussion section:
“No sexual orientation differences emerged between the non-vegetarian group and vegetarian/semi-vegetarian adolescents. These statistically non-significant results may be due to the small sample sizes of the groups. Moreover, as highlighted elsewhere [48], the link between diet patterns and sexual orientation is complex, and studies are still lacking. Therefore, further research is needed to consider identity development processes in adolescence.”
20.
“Your attitudes towards nutrition:”
Delete Yours. Always use small (non-capital) letters when not starting a sentence.
Answer 20: The authors prefer not to delete Yours as in the original questionnaire [44]. The following note has been added: “The original heading of this section of the questionnaire by Lea and Worsley [52] was “Your attitude towards nutrition and meat”. Here it has been simplified by omitting “meat” to introduce the measures selected for the topic of the current study.”
- Add all the items in the supplementary material for the interested reader.
Answer 21: Items are available in the supplementary material. Appendix A
- “(from 1 = strongly disagree to 5 = strongly agree”
Correct capitalization and inconsistent spacing (in the whole paper).
Answer 22. Thanks, we revised the whole text.
23.“Over the past six months, how low have you felt about certain events?”. Response varies on a 4-point Likert scale (from 1= = not agree, to 4=agree a lot).”
What were the depression items? Please, put in the supplementary material.
Answer 23: Items are available in the supplementary material. Appendix A
- A reader would welcome more concrete policy recommendations stemming from the study findings.
Answer 25. Thank you for the suggestions. A new section with concrete implications was added.
- Implications
Understanding favourable factors/barriers to adopting a vegetarian diet or reducing meat consumption (as in semi-vegetarians' case) offers an essential insight for planning interventions. Health professionals (i.e., paediatricians, nutritionists), and other significant adults (such as teachers or sports trainers), should respect the dietary choices of adolescents and their families. This involves understanding the cultural, ethical or religious motivations influencing everyone's eating styles and not considering a plant-based diet as deviant or eccentric. In addition, health professionals should be guided by empirically based information about healthy vegetarian (or vegan) diets. Vegetarian adolescents generally know the health benefits of a meat-free diet. Still, they may fear not knowing how to manage it adequately, balancing the intake of nutrients contained in meat and derivatives. To promote a well-balanced vegetarian diet and prevent nutritional deficiencies, dietary-nutritional consultations could be offered to accompany adolescents in their decision-making process under the guidance of experts [38], [40].
Community interventions (i.e., school-based) can effectively disseminate correct nutritional education. However, it is known that increasing knowledge (i.e., providing information on the benefits of a plant-based diet by nutritionists) is not an intervention that guarantees the adoption of healthy behaviours among adolescents [54]. Conversely, practical strategies for promoting a healthy diet could be implemented in educational programs. A study [61] explored factors affecting the preference for vegetables among high school students. Regardless of the adolescents’ weight status (BMI), the personal factors determinant for increasing the vegetable intake resulted in positive attitudes toward plant-based foods, and self-efficacy, that is, adolescents’ confidence to make their own choices and achieve personal goals. Among environmental factors, vegetable accessibility and positive experiences with plant-based foods resulted in determinants for adolescents’ choices. These findings suggest the importance of supporting adolescents’ positive attitudes and experiences with plant-based foods and improving personal skills (such as self-efficacy and decision-making). Interestingly, learning to cook vegetarian foods and discovering that they are appetising can also increase adolescents' life skills (e.g., self-efficacy, perception of control over health etc.) and psychological well-being [56].
Similarly, active interventions (such as focus-group [55]) can effectively offer participants opportunities to openly express their beliefs about healthy nutrition or to share positive experiences that can motivate a shift in dietary patterns [62]. Education might also modify adolescents’ barriers or misconceptions regarding a plant-based diet, for example, parental influences or the expectation of the tastiness of vegetable foods compared with a good flavour of the meat [56].
Finally, the practical implications of the current study derive from the weight attributed by vegetarian adolescents (less by semi-vegetarians) to ethical motivations, particularly the sustainability and ecological benefits of vegetarian/vegan diets. The issues of the low environmental impact of plant-based foods (against damage of intensive animal breeding for pollution and climate change) are becoming popular among young people [18]. Values associated with a vegetarian lifestyle are social equity and accessibility of alternatives to meet among most people worldwide against hunger and inequality. Future research based on community qualitative (or mixed) methods could explore these emerging ethical issues, with the scope to understand their weight in driving adolescents' dietary decisions and guide policies and interventions. For example, we suggest that future researchers use Online Photovoice (OPV)[63], [64] to research the same or similar topics. It is an innovative qualitative research method that allows the participants to express their own experiences with as little manipulation as possible, compared to traditional quantitative methods.
26.
“Differently from the adult population, where vegetarianism is a predominantly female phenomenon [12], there were no significant differences in gender between the non-vegetarian group and vegetarian/semi-vegetarian adolescents.”
Readers would welcome a discussion on potential reasons for the lack of gender differences.
Answer 26. the authors thank the anonymous reviewer for this observation, which allowed them to notice an error in the reading of the data. In fact, the statistical value of comparison among the three groups has been commented on (table 1): the non-significant value of the chi-square is expected, since it confirms that the groups (vegetarians, semi-vegetarians, and no vegetarians reduced group n= 40) are equivalent.
The chi-square was then calculated on the entire sample, and it resulted in significant. As a consequence, the text has been modified in two places as follows:
A] comment on the results (paragraph 4.1):
4.1. Prevalence of vegetarian diet and gender differences
Within the total sample of participants (n=1019), 93.7% declared themselves non-vegetarian (n=962), 43.9% male (n=447) and 50.5% female (n=515). Participants who defined themselves as semi-vegetarian (n=39) – they occasionally eat foods of animal origin – were 3.9% representing the choice of 1.6% males (n= 16) and 2.3% females (n= 23). Finally, 4.5% declared themselves vegetarian (n= 18), corresponding to 3% of males (n= 3) and 1.5% of females (n= 15) of the total sample.
A chi-square test applied to a cross tab 2 (gender) x 3 (groups: NV, V, and SV) revealed the non-independence between factors [χ²(2) = 6.68, p ‹ .05].
B] Discussion
“In line with previous studies with the adult population, where vegetarianism is a more common dietary habit in women, adolescents who self-identify as vegetarian/semi-vegetarian are more likely to be females. Therefore, current data follow the growing interest in gender differences in dietary choice by enriching findings on the adolescent population [12].”
- “These findings confirm that the vegetarian choice among adolescents is strongly sustained by adopting a healthy approach to life.”
Vague, unclear.
Answer 28. As suggested, the sentence has been corrected.
“These findings confirm that the vegetarian choice among adolescents is strongly sustained by adopting a healthy approach to diet.”
- “This is also confirmed by the results of the sections Barriers and Benefits, in which youths provided feedback on how they perceive (“good” or “problematic”) the vegetarian diet approach. Particularly, in the measure of benefits, they described their beliefs about health-positive aspects (such as the prevention of disease, e.g. heart disease, cancer), personal aspects (i.e. being more content with themself, satisfying religious and/or spiritual needs), and ethic instances (such as the respects of the environment, the fight against inequalities, or animal welfare/rights). In the section Health (alpha= .92 in the current study), vegetarians and semi-vegetarians reported higher scores than non-vegetarians. Similarly, in the section Personal benefits (alpha=. 83), vegetarians and semi-vegetarians significantly differ from non-vegetarians who reported lower scores. Finally, in the questions about Ethical benefits (alpha= .84),”
Alphas do not belong in the Discussion sections but in the Method section. Second, capitalization. Third, clearly state what construct was measured, not stating vague names of the section. Fourth, summarize key results, and be concrete. Goes for the next paragraphs as well.
Answer 29. As requested, alphas (It. adaptation of the questionnaire) have been moved to the method section (3.3 Measures). A more detailed description of the constructs assessed by the questionnaire [44] was also added.
The changes are the following:
“Food choice, information and your attitudes” is a self-report questionnaire investigating several individual nutrition-related factors. The questionnaire was adapted to the Italian language from a previous study with adolescents [51] with the authors' permission. For the purpose of the current study, the selected sections are the following.
“Your attitudes towards nutrition”. The original heading of this section of the questionnaire by Lea and Worsley [52] was “Your attitude towards nutrition and meat”. Here it has been simplified by omitting “meat” to introduce the measures selected for the topic of the current study. The first section (4 items) assesses attitudes towards nutrition, for example, “I frequently look for information on healthy eating”. The answers are on a 5-point Likert scale (1=strongly disagree to 5=strongly agree). This section explores the knowledge of diet in preventing illness and disease or the attention related to the nutritional aspect of the type of food usually consumed, or again the ideas about healthy eating. A specific question asks: “Are you vegetarian now?”. Respondents can select one of three answers:(1)no, (2)yes, or (3)“I normally think of myself as being a semi-vegetarian”. This item was used to classify the students according to their self-identified dietary style and to submit the results for statistical analysis. The other two items explored family and peer factors: “Do you have any vegetarian family members? (None/father/mother/sibling/s/partner) and “Do you have any vegetarian friends? (None/a few/about a quarter of my friends/about half of my friends/about three quarter of my friends/all).
Barriers to vegetarian diets. This section 21 items in the original questionnaire [51] investigates how difficult it could be to be a vegetarian (for example: “I would (or do) feel conspicuous among others”, “There is not enough iron in vegetarian diets”). The agreement is expressed on a 5-point Likert scale (from 1= strongly disagree to 5= strongly agree). In the Italian adaptation [51], the assessed issues are dietary difficulties (5 items, alpha = .76), that evaluate problems with a restricted food selection (e.g., “Vegetarian diets are not filling enough”); health-related concerns (5 items, alpha = .79) regarding nutritional deficiencies (proteins, calcium etc.) resulting from plant-based foods (e.g. “There is not enough protein in vegetarian diets”); food family/friends habits (2 items, alpha = .85), that is, “my friends/my family eat(s) meats”).
Benefits of vegetarian diets. A list of statements (24 items) evaluates the perceived benefits of a vegetarian diet (“I believe a vegetarian diet could (or does) help me to…”). Examples of items are: “control my weight” or “create a more peaceful world”. The respondent rates his/her agreement on each item using a 5-point Likert scale (from 1= strongly disagree to 5= strongly agree). Three measurements are obtained [51]: health benefits (12 items, alpha =.92), that is, preventing diseases in general (e.g. heart disease, cancer) and promoting well-being (e.g. “increase my control over my health”); personal benefits (6 items, alpha = .83) both physical and psychological (“have plenty of energy” and “satisfying religious and/or spiritual needs”); ethical benefits such as animal welfare, social equity or environmental reasons (e.g. “the efficiency of food production”; 6 items, alpha=.84).
In addition, the discussion section has been modified as suggested by the referee: a) by removing the capital letters where present, b) with a summary of the results and a comment integrating data of the current study with literature.
This is also confirmed by the results of the sections barriers and benefits, in which youths provided feedback on how they perceive (“good” or “problematic”) the vegetarian diet approach. Notably, in the measure of benefits, they described their beliefs about health-positive aspects (e.g., the prevention of disease such as heart disease or cancer), personal aspects (i.e., being more content with themself, satisfying religious and/or spiritual needs), and ethic instances (such as the respect of the environment, the fight against inequalities, or animal welfare/rights). In the section health, vegetarians and semi-vegetarians reported higher scores than non-vegetarians, and the adherence to health reasons did not differentiate teenagers who define themselves as strict vegetarians from semi-vegetarians. Adolescents who choose a vegetarian diet seem aware of the positive health consequences of excluding meat intake. Conversely, non-vegetarians share inadequate or superficial knowledge about the health benefits of vegetarianism, as observed in a previous study [55].
In the section personal benefits vegetarians and semi-vegetarians provided similar answers, and both groups significantly differed from non-vegetarians who reported the lower scores. These beliefs include a positive perception of vegetarian diets, such as food taste or energy value. In particular, the (expected) poor taste of plant-based foods, together with proximal factors (i.e., family support and diet), is one of the reasons declared by adolescents as little interested in introducing more plant-based foods[56]. In the current study, the assessed personal benefits include psychological correlates such as coherence with spiritual/religious values, self-esteem, and satisfaction with one's choices. These perceived benefits and defining oneself as a "vegetarian" satisfy psychological needs crucial for an adolescent’s identity and well-being.
Finally, in the questions about ethical benefits, vegetarian adolescents reported higher scores than non-vegetarians and semi-vegetarians. These results suggest that a constellation of values oriented towards justice, nature, the promotion of a peaceful world, and respect for differences are most felt by vegetarian adolescents compared to semi-vegetarian adolescents. Alongside the already-known themes of animal rights and the refusal of any cause of suffering to other beings, an interesting ethical issue that emerges in the opinions of adolescents (mainly vegetarians) is the environmental impact and sustainability of vegetarian foods (e.g., less pollution, more efficient food production, or fight against hunger). This aspect is little investigated among vegetarian adolescents [56], despite the growing sensitivity of young people towards environmental issues. Furthermore, findings from the current study have the advantage of increasing knowledge about the reasons underlying adolescents' vegetarian lifestyle, but further studies are needed.
The section barriers assessed adolescents’ beliefs about why it may be challenging to be a vegetarian: dietary difficulties include motives such as a limited choice/energy from meat-free foods or the taste of meat; health-related concerns measure worries about nutritional deficiencies of the vegetarian diet, e.g. insufficient balance of protein and/or micronutrients; family/friends' habits describe worries about diversity of dietary choice (mainly eating meat) among peers and/or family members.
- “Non-vegetarian adolescents utter the greatest difficulties due to dietary restrictions and health-related issues.”
Change “utter” to “report”.
Answer 31. Thanks for the correction. We replaced the word.
32.“authors firmly believe”
Please, delete this phrasing.
Answer 32. Thanks for the note, eliminated.
33.“Institutional Review Board Statement: The study was conducted following the Declaration of Helsinki, and the ethical standard norms for psychological research of the Italian Psychology Association (AIP; chrome-extension://efaidnbmnnnibpcajpcglclefindmkaj/https://aipass.org/wp-content/uploads/2023/02/Codice-Etico_luglio-2022.pdf).”
The link states Chrome extensions? Please, correct.
Answer 33. Thanks for letting us know, it was a typo. We corrected.
34.“Data Availability Statement: The data presented in this study are available in the section 3. Materials and Methods, and 4. Results in the article.”
It should be stated that survey data is available from the first (or lead) author upon reasonable request (please, see other MDPI papers).
Answer 34. Thanks, we corrected in “The study was conducted following the Declaration of Helsinki, and was approved by Ethics Committee of the Psychological Research and Intervention Center (CeRIP), University of Messina. Approval Code: n. 0032071/a [UOR: SI001165-Classif. III/11]. Approval Date: 08/03/2023.”
35.““C. Larsson, “Young vegetarians and omnivores Dietary habits and other health-related aspects”.”
Please, provide all the needed information of all references in the references list.
Answer 35. We provide to revise the reference list. In the text, you will find the corrections highlighted in yellow.
- Considering the above-stated comments, I suggest the authors revise the paper.
Answer 36. Thanks for all the suggestions and corrections. They were punctual, valuable and accurate. We tried to solve the different issues by following the guide.
I hope you are convinced that we have adequately addressed the reviewer’s concerns and improved the paper.
Reviewer 2 Report
Comments and Suggestions for Authors
Title of the work: Teenagers' beliefs and well-being in dietary decisions motivate them to choose a Vegetarian lifestyle.
Dear researcher(s), you are addressing an important and meaningful gap. Your paper is well-written and it has some important results, and if you edit your paper it can be much more effective. Here some humble suggestions to improve the paper, I would do the following to strengthen the paper. I have enjoyed reading the paper and am looking forward to seeing the paper published. You could increase the effect of your paper with some more recent studies suggested below or any other studies and not using the suggested ones. I have enjoyed reading the paper and please see my attached feedback.
I humbly invite you to respond to my comments and revise your manuscript. When you revise your manuscript please highlight the changes you make in the manuscript by using the track changes mode in Word or by using coloured text; I would prefer coloured text. In this way I can see the changes much quicker.
Main points:
1. Title: good
2. Abstract and keywords clear and comprehensive: very good
3. Overall language:
- The language is quite clear and well-written. You could use an active language for your future papers throughout the paper since an active language seems to be more effective. And more and more researchers go with an active language. However, you do not have to change for this paper- just a suggestion for your future work and I know some journals asking for a passive language.
- For an active language see suggested papers
4. Length of paragraph : good and
a. you can check the paper and make sure every paragraph is not more than 5 sentences. The best is to stick with 3 to 5 sentences.
b. And you can add some subtitles to be more organized and short sections. Please see the suggested papers above. I would suggest you to look at the suggested studies and use more subtitles to make the paper more organized.
5. Introduction: good and can use subtitles to make it more organized to follow subjects
6. Thoroughness of the literature review: can support with some recent studies yet do not have to. This will increase the effect of the paper and the journal.
7. Clarity of the description of the Theoretical Framework (TF): You do not have to construct a subsection called theoretical framework because you have enough explanation about which theories/perspectives have shaped your research. However, if you want to make your paper more effective I would construct a subsection called theoretical framework and briefly explains see suggested papers
8. Research design: very good
9. Clearly providing research questions and/or purpose: very good
10. Method: Choice of research method: very appropriate
11. Appropriateness of procedures chosen for data collection and analysis: well-written
12. Results: Relevance of data obtained in view of the purpose of the research: well-written
13. Discussion of the results and their significance: well-written
14. Soundness of conclusions in relation to data presented: well-written.
15. Limitation: good and can be listed under a clear limitation/weaknesses serction
16. Implication:
a. you can increase the effect of your paper by constructing a new section entitled “implication” for clear and brief suggestions in at least two or three of the following most important to you mental health, education, research, administrators, services, etc.: see suggested papers for implications for specific sections
b. you can call for future researchers to use a community based approach and act with and for people dealing with this subject for more grounded research and services, See suggested studies
c. I would strongly suggest you to call future researchers to use Online Photovoice (OPV) to conduct research on the same or similar topics. The researchers can use OPV, as one of the most recent and effective innovative qualitative research methods. OPV gives opportunities to the participants to express their own experience with as little manipulation as possible if at all, compared to traditional quantitative methods. As researchers one of our responsibilities is to inform others about recent and effective methods, which will increase the effect of your paper and the journal. Future researchers can conduct only qualitative or mixed method to see if OPV. And educators/trainers etc. also can use OPV for experiential activities to increase group and organizational synergy. Please see suggested papers if you wish to do so.
DoyumÄŸaç, İ., Tanhan, A., & Kıymaz, M. S., (2021). Understanding the most important facilitators and barriers for online education during COVID-19 through online photovoice methodology. International Journal of Higher Education, 10(1), 166-190. https://doi.org/10.5430/ijhe.v10n1p166
https://scholar.google.com/scholar?hl=en&as_sdt=0%2C5&q=Understanding+the+most+important++facilitators+and+barriers+for+online+education+during+COVID-19+through+online+photovoice+methodology&btnG=
Tanhan, A., & Strack, R. W. (2020). Online photovoice to explore and advocate for Muslim biopsychosocial spiritual wellbeing and issues: Ecological systems theory and ally development. Current Psychology, 39(6), 2010-2025. https://doi.org/10.1007/s12144-020-00692-6
https://scholar.google.com/scholar?hl=en&as_sdt=0%2C5&q=Online+photovoice+to+explore+and+advocate+for+Muslim++biopsychosocial+spiritual+wellbeing+and+issues%3A+Ecological+systems+theory+and+ally+development&btnG=
Dari, T., Fox, C., Laux, J. M., & Speedlin Gonzalez, S. (2023). The Development and Validation of the Community-Based Participatory Research Knowledge Self-Assessment Scale (CBPR-KSAS): A Rasch Analysis. Measurement and Evaluation in Counseling and Development, 56(1), 64-79. https://doi.org/10.1080/07481756.2022.2034478
Hauber-Özer, M., Call-Cummings, M., Hassell-Goodman, S., & Chan, E. (2021). Counter- storytelling: Toward a critical race praxis for participatory action research. International Journal of Qualitative Studies in Education. https://doi.org/10.1080/09518398.2021.1930252
17. Figure/tables: good
18. In-text reference: good
19. References: good
- Please use the following link to include all available doi numbers if the journal allows https://doi.crossref.org/simpleTextQuery simply include your reference one or more than one at a time and submit it. Then you should get all doi numbers if a manuscript has it.
I have enjoyed reading your paper and learned a lot- thanks for your contribution to social sciences. You are addressing an important and meaningful gap. Your paper has some important results, and if you edit your paper based on all or some of the humble suggestions above, it can be much more effective. I am looking forward to seeing the paper published. You could increase the effect of your paper with some more recent studies suggested above.
Author Response
Reviewer 2
Dear researcher(s), you are addressing an important and meaningful gap. Your paper is well-written and it has some important results, and if you edit your paper it can be much more effective. Here some humble suggestions to improve the paper, I would do the following to strengthen the paper. I have enjoyed reading the paper and am looking forward to seeing the paper published. You could increase the effect of your paper with some more recent studies suggested below or any other studies and not using the suggested ones. I have enjoyed reading the paper and please see my attached feedback. I humbly invite you to respond to my comments and revise your manuscript. When you revise your manuscript please highlight the changes you make in the manuscript by using the track changes mode in Word or by using coloured text; I would prefer coloured text. In this way I can see the changes much quicker.
Main points:
- Title: good
- Abstract and keywords clear and comprehensive: very good
- Overall language:
- The language is quite clear and well-written. You could use an active language for your future papers throughout the paper since an active language seems to be more effective. And more and more researchers go with an active language. However, you do not have to change for this paper- just a suggestion for your future work and I know some journals asking for a passive language.
- For an active language see suggested papers
Answer 3. We appreciate your advice and will consider using more direct language for future articles. It would give the idea of being more effective and immediate. Thanks for the suggestion; we’ll read your suggested papers.
- Length of paragraph : good and
- you can check the paper and make sure every paragraph is not more than 5 sentences. The best is to stick with 3 to 5 sentences.
- And you can add some subtitles to be more organized and short sections. Please see the suggested papers above. I would suggest you to look at the suggested studies and use more subtitles to make the paper more organized.
Answer 4. Thanks for the comment, we have remodulated the length of some paragraphs.
- Introduction: good and can use subtitles to make it more organized to follow subjects
- Thoroughness of the literature review: can support with some recent studies yet do not have to. This will increase the effect of the paper and the journal.
- Clarity of the description of the Theoretical Framework (TF): You do not have to construct a subsection called theoretical framework because you have enough explanation about which theories/perspectives have shaped your research. However, if you want to make your paper more effective I would construct a subsection called theoretical framework and briefly explains see suggested papers.
Answer 7. Thanks for the observation. We added the following description of the TF.
“Most studies investigating factors impacting adolescents' food choices describe multiple factors that operate on multiple levels of influence [25]. Assuming Bronfenbrenner’s (1979) [26] ecological perspective, the factors can be described at different interacting levels: individual, including biological (e.g., weight) and psychological factors (e.g., attitudes, taste preferences, perceived barriers etc.); interpersonal, such as family or peer influences (through modelling or perceived support); environmental (e.g., availability and cost of foods, etc.), cultural or macrosystem factors that play an indirect influence on food behaviours (e.g., mass media, religion or societal norms of eating).”
- Research design: very good
- Clearly providing research questions and/or purpose: very good
- Method: Choice of research method: very appropriate
- Appropriateness of procedures chosen for data collection and analysis: well-written
- Results: Relevance of data obtained in view of the purpose of the research: well-written
- Discussion of the results and their significance: well-written
- Soundness of conclusions in relation to data presented: well-written.
- Limitation: good and can be listed under a clear limitation/weaknesses section
- Implication:
- you can increase the effect of your paper by constructing a new section entitled “implication” for clear and brief suggestions in at least two or three of the following most important to you mental health, education, research, administrators, services, etc.: see suggested papers for implications for specific sections
- you can call for future researchers to use a community based approach and act with and for people dealing with this subject for more grounded research and services, See suggested studies
- I would strongly suggest you to call future researchers to use Online Photovoice (OPV) to conduct research on the same or similar topics. The researchers can use OPV, as one of the most recent and effective innovative qualitative research methods. OPV gives opportunities to the participants to express their own experience with as little manipulation as possible if at all, compared to traditional quantitative methods. As researchers one of our responsibilities is to inform others about recent and effective methods, which will increase the effect of your paper and the journal. Future researchers can conduct only qualitative or mixed method to see if OPV. And educators/trainers etc. also can use OPV for experiential activities to increase group and organizational synergy. Please see suggested papers if you wish to do so.
Answer 16. Thanks for this suggestion. A new section (“6. Implication”) was added with suggestions for future qualitative studies.
- Implications
Understanding favourable factors/barriers to adopting a vegetarian diet or reducing meat consumption (as in semi-vegetarians' case) offers an essential insight for planning interventions. Health professionals (i.e., paediatricians, nutritionists), and other significant adults (such as teachers or sports trainers), should respect the dietary choices of adolescents and their families. This involves understanding the cultural, ethical or religious motivations influencing everyone's eating styles and not considering a plant-based diet as deviant or eccentric. In addition, health professionals should be guided by empirically based information about healthy vegetarian (or vegan) diets. Vegetarian adolescents generally know the health benefits of a meat-free diet. Still, they may fear not knowing how to manage it adequately, balancing the intake of nutrients contained in meat and derivatives. To promote a well-balanced vegetarian diet and prevent nutritional deficiencies, dietary-nutritional consultations could be offered to accompany adolescents in their decision-making process under the guidance of experts [38], [40].
Community interventions (i.e., school-based) can effectively disseminate correct nutritional education. However, it is known that increasing knowledge (i.e., providing information on the benefits of a plant-based diet by nutritionists) is not an intervention that guarantees the adoption of healthy behaviours among adolescents [54]. Conversely, practical strategies for promoting a healthy diet could be implemented in educational programs. A study [61] explored factors affecting the preference for vegetables among high school students. Regardless of the adolescents’ weight status (BMI), the personal factors determinant for increasing the vegetable intake resulted in positive attitudes toward plant-based foods, and self-efficacy, that is, adolescents’ confidence to make their own choices and achieve personal goals. Among environmental factors, vegetable accessibility and positive experiences with plant-based foods resulted in determinants for adolescents’ choices. These findings suggest the importance of supporting adolescents’ positive attitudes and experiences with plant-based foods and improving personal skills (such as self-efficacy and decision-making). Interestingly, learning to cook vegetarian foods and discovering that they are appetising can also increase adolescents' life skills (e.g., self-efficacy, perception of control over health etc.) and psychological well-being [56].
Similarly, active interventions (such as focus-group [55]) can effectively offer participants opportunities to openly express their beliefs about healthy nutrition or to share positive experiences that can motivate a shift in dietary patterns [62]. Education might also modify adolescents’ barriers or misconceptions regarding a plant-based diet, for example, parental influences or the expectation of the tastiness of vegetable foods compared with a good flavour of the meat [56].
Finally, the practical implications of the current study derive from the weight attributed by vegetarian adolescents (less by semi-vegetarians) to ethical motivations, particularly the sustainability and ecological benefits of vegetarian/vegan diets. The issues of the low environmental impact of plant-based foods (against damage of intensive animal breeding for pollution and climate change) are becoming popular among young people [18]. Values associated with a vegetarian lifestyle are social equity and accessibility of alternatives to meet among most people worldwide against hunger and inequality. Future research based on community qualitative (or mixed) methods could explore these emerging ethical issues, with the scope to understand their weight in driving adolescents' dietary decisions and guide policies and interventions. For example, we suggest that future researchers use Online Photovoice (OPV)[63], [64] to research the same or similar topics. It is an innovative qualitative research method that allows the participants to express their own experiences with as little manipulation as possible, compared to traditional quantitative methods.
DoyumÄŸaç, İ., Tanhan, A., & Kıymaz, M. S., (2021). Understanding the most important facilitators and barriers for online education during COVID-19 through online photovoice methodology. International Journal of Higher Education, 10(1), 166-190. https://doi.org/10.5430/ijhe.v10n1p166
https://scholar.google.com/scholar?hl=en&as_sdt=0%2C5&q=Understanding+the+most+important++facilitators+and+barriers+for+online+education+during+COVID-19+through+online+photovoice+methodology&btnG=
Tanhan, A., & Strack, R. W. (2020). Online photovoice to explore and advocate for Muslim biopsychosocial spiritual wellbeing and issues: Ecological systems theory and ally development. Current Psychology, 39(6), 2010-2025. https://doi.org/10.1007/s12144-020-00692-6
https://scholar.google.com/scholar?hl=en&as_sdt=0%2C5&q=Online+photovoice+to+explore+and+advocate+for+Muslim++biopsychosocial+spiritual+wellbeing+and+issues%3A+Ecological+systems+theory+and+ally+development&btnG=
Dari, T., Fox, C., Laux, J. M., & Speedlin Gonzalez, S. (2023). The Development and Validation of the Community-Based Participatory Research Knowledge Self-Assessment Scale (CBPR-KSAS): A Rasch Analysis. Measurement and Evaluation in Counseling and Development, 56(1), 64-79. https://doi.org/10.1080/07481756.2022.2034478
Hauber-Özer, M., Call-Cummings, M., Hassell-Goodman, S., & Chan, E. (2021). Counter- storytelling: Toward a critical race praxis for participatory action research. International Journal of Qualitative Studies in Education. https://doi.org/10.1080/09518398.2021.1930252
Due to the introduction of the “Implications” section, the conclusions are re-numbered 7 and some repetitions are revised as follows:
- Conclusions
The current study explored how vegetarianism among adolescents aligns with various health, ethics, and environmental sustainability values. The findings indicate that ethical considerations primarily motivate adolescents who self-identify as “strict” vegetarians (compared with semi-vegetarians), while the perceived health benefits are considered additional advantages. Among individual factors associated with vegetarianism, a predominance of girls and a low adherence to traditional religions resulted. Vegetarian adolescents more often come from families where others follow meat-free diets. Regarding well-being indicators, vegetarianism is not associated with adolescents’ desire for weight loss or body dissatisfaction.
Gaining autonomy and making choices consistent with one's goals and values are critical developmental tasks in adolescence. Since being self-determined increases personal well-being and can lead to better neuropsychological development [65]–[66], it is essential to support adolescents’ dietary choices and provide information about a balanced diet, the correct vitamin reintegration and dietary protein intake, as the scientific community recommends.
Interventions should prevent health and nutritional problems associated with unbalanced dietary choices but also support adolescents’ dietary lifestyle as a component of a more complex view of the world oriented towards ethical values, animal welfare, fight against inequalities, and defence of the environment.
- Figure/tables: good
- In-text reference: good
- References: good
- Please use the following link to include all available doi numbers if the journal allows https://doi.crossref.org/simpleTextQuery simply include your reference one or more than one at a time and submit it. Then you should get all doi numbers if a manuscript has it.
Answer 19. Thanks for the helpful and constructive suggestion. We searched for suggested items and found them very interesting. Surely we will use them as a guide for writing our next works.
I have enjoyed reading your paper and learned a lot- thanks for your contribution to social sciences. You are addressing an important and meaningful gap. Your paper has some important results, and if you edit your paper based on all or some of the humble suggestions above, it can be much more effective. I am looking forward to seeing the paper published. You could increase the effect of your paper with some more recent studies suggested above.
Reviewer 3 Report
Comments and Suggestions for Authors
This article explores how vegetarianism aligns with various values such as health, ethics, and environmental sustainability, particularly among adolescents. The research focuses on factors influencing this dietary choice, including gender, religious and ethical beliefs, and well-being indicators. The findings indicate that ethical considerations primarily motivate adolescent vegetarianism, while health benefits are viewed as supplementary advantages. The choice to become vegetarian is not associated with desires for weight loss or body dissatisfaction.
While the topic is compelling, the small sample size limits the ability to draw conclusive results from this data. Despite its importance, this study could benefit from a research strategy that involves a larger sample of vegetarians or a broader population.
Moreover, the article lacks detailed information about the statistical methods employed. Please fully spell out acronyms like ANOVA and MANOVA upon their first mention and specify which variables were adjusted for in the MANOVA analysis. Additionally, information about the statistical software used and the level of significance considered should be included.
You assert in your discussion that "the very large non-clinical sample of Italian adolescents enrolled (n=1040) allows for robust conclusions." However, this sample size may not be large enough to be accurately termed as "very large."
Lastly, has your research been subjected to an ethical review by your institution? The article does not provide any information about the name or approval number of the Ethical Review Board.
Author Response
Reviewer 3
This article explores how vegetarianism aligns with various values such as health, ethics, and environmental sustainability, particularly among adolescents. The research focuses on factors influencing this dietary choice, including gender, religious and ethical beliefs, and well-being indicators. The findings indicate that ethical considerations primarily motivate adolescent vegetarianism, while health benefits are viewed as supplementary advantages. The choice to become vegetarian is not associated with desires for weight loss or body dissatisfaction.
While the topic is compelling, the small sample size limits the ability to draw conclusive results from this data. Despite its importance, this study could benefit from a research strategy that involves a larger sample of vegetarians or a broader population.
Moreover, the article lacks detailed information about the statistical methods employed. Please fully spell out acronyms like ANOVA and MANOVA upon their first mention and specify which variables were adjusted for in the MANOVA analysis. Additionally, information about the statistical software used and the level of significance considered should be included.
Answer: Thank you very much for the comments and suggestions. Acronyms were fully spelt; information about the statistical software used and the level of significance considered were added. No variables were adjusted for the MANOVA: the random extraction of participants from the total NV group considered only gender and age as matching characteristics with the other groups. We modifyed the paragraph as follows:
3.4 Statistical analysis
Based on self-reported dietary style, the sample was divided into three groups, that is, non-vegetarians (NV), vegetarians (V), and semi-vegetarians (SV). The significance of differences in gender proportions among the three groups was tested by a contingency table 2 (male vs. female gender) x 3 (NV, V, and SV dietary style) and the chi-square test. Subsequently, a subgroup of participants was randomly selected from the broader non-vegetarian sample, matching them by age and gender proportion (16 males and 24 females) to the vegetarian and semi-vegetarian subgroups. Therefore, a series of crosstab chi-square tests tested the independence of dietary patterns with sociodemographic factors (i.e., gender, sexual orientation, religion) and weight-related factors (i.e. BMI categories and weight satisfaction rates). Parametric variables according to dietary patterns were compared using F tests.
Finally, group differences (NV, V, and SV) were estimated for all study measures: by one-way analysis of variance (ANOVA) for food attitudes; and by multivariate analysis of variance (MANOVA), or non-parametric median test when variances were not homogeneous on Levene’s test, for benefits/difficulties with vegetarian diet.
All analyses were performed using IBM SPSS Statistics 22. For all tests the p<.05 significance level was established.
You assert in your discussion that "the very large non-clinical sample of Italian adolescents enrolled (n=1040) allows for robust conclusions." However, this sample size may not be large enough to be accurately termed as "very large."
Lastly, has your research been subjected to an ethical review by your institution? The article does not provide any information about the name or approval number of the Ethical Review Board.
Answer: Thanks for your comment.
We changed the incorrect term “very large non-clinical sample” to “our” in the test.
We provide information about the name or approval number of the Ethical Review Board in the apposite section “Institutional Review Board Statement”, how we report hereafter: “The study was conducted following the Declaration of Helsinki, and was approved by Ethics Committee of the Psychological Research and Intervention Center (CeRIP), University of Messina. Approval Code: n. 0032071/a [UOR: SI001165-Classif. III/11]. Approval Date: 08/03/2023.”

Round 2
Reviewer 3 Report
Comments and Suggestions for Authors
The issue of sample size has not been resolved. Is there a basis for conducting research with this sample size? It would be preferable to have a method for determining the sample size described.
I have no other comments, but as for comments from reviewers, it is preferable to respond to each comment individually.
Author Response
Thank you for the comment. We apologize if we have not responded exhaustively to each comment individually.
We agree with the observation: it is usually preferable to have a method for determining the size of the recruited sample.
When we started this exploratory study, we knew that the school population of Catanzaro was 16354 students distributed in 466 classrooms. Therefore, we decided to recruit about a tenth of the students, representing all types of secondary schools in Catanzaro, to investigate the incidence of vegetarian adolescents. Only after having identified vegetarians, semi-vegetarians, and non-vegetarians, the characteristics that differentiated the groups were investigated on specific incidence numbers. An a priori method, as the Andrew Fisher’s formula, for recruiting vegetarian people in Catanzaro schools was not available because the incidence of this group in that city was unknown.
For these reasons, we will not change the part of the article that describes the sampling. However, in the study's limitations, the participants' local characteristics are discussed.